# Improving Rural Healthcare in Mobile Clinics: Real-Time, Live Data Entry into the Electronic Medical Record Using a Satellite Internet Connection

**DOI:** 10.3390/ijerph22060842

**Published:** 2025-05-28

**Authors:** Daniel Jackson Smith, Elizabeth Mizelle, Nina Ali, Valery Cepeda, Tonya Pearson, Kayla Crumbley, Dayana Pimentel, Simón Herrera Suarez, Kenneth Mueller, Quyen Phan, Erin P. Ferranti, Lori A. Modly

**Affiliations:** 1School of Nursing, University at Buffalo, Buffalo, NY 14214, USA; ninaali@buffalo.edu; 2Farmworker Family Health Program, Atlanta, GA 30322, USA; mizelleel15@ecu.edu (E.M.); valerycepeda0422@gmail.com (V.C.); pearson_t@mercer.edu (T.P.); rogers.kayla501@gmail.com (K.C.); dayanapim.main@gmail.com (D.P.); simonherrera123@gmail.com (S.H.S.); kjmuell@emory.edu (K.M.); qphan@emory.edu (Q.P.); epoe@emory.edu (E.P.F.); 3College of Nursing, East Carolina University, Greenville, NC 27858, USA; 4School of Pharmacy, University of Georgia, Athens, GA 30602, USA; 5College of Pharmacy, Mercer University, Atlanta, GA 30341, USA; 6Nell Hodgson Woodruff School of Nursing, Emory University, Atlanta, GA 30322, USA; 7Rollins School of Public Health, Emory University, Atlanta, GA 30322, USA

**Keywords:** farmworker health, real-time data entry, satellite internet, healthcare accessibility, mobile clinics

## Abstract

The Farmworker Family Health Program (FWFHP) annually supports 600 farmworkers in connectivity-challenged rural areas. Traditional paper-based data collection poses validity concerns, prompting a pilot of direct data entry using tablets and satellite internet to enhance efficiency. The purpose of this article is to describe, using the TIDier checklist, a real-time, live data-entry EMR intervention made possible by satellite internet. Utilizing a customized REDCap database, direct data entry occurred through tablets and satellite internet. Patients received a unique medical record number (MRN) at the mobile health clinic, with an interprofessional team providing care. Medication data, captured in REDCap before the mobile pharmacy visit, exhibited minimal defects at 6.9% of 319 prescriptions. To enhance data collection efficiency, strategies such as limiting free text variables and pre-selecting options were employed. Adequate infrastructure, including tablets with keyboards and barcode scanners, ensured seamless data capture. Wi-Fi extenders improved connectivity in open areas, while backup paper forms were crucial during connectivity disruptions. These practices contributed to enhanced data accuracy. Real-time data entry in connectivity-limited settings is viable. Replacing paper-based methods streamlines healthcare provision, allowing timely collection of occupational and environmental health metrics. The initiative stands as a scalable model for healthcare accessibility, addressing unique challenges in vulnerable communities.

## 1. Introduction

Health information technology (HIT) has the potential to improve healthcare access, outcomes, and quality [1]. However, due to persistent socioeconomic and geographic disparities, its benefits remain uneven. A digital divide driven by limited technology access and internet use [2] continues to affect underserved populations, particularly along racial, economic, and rural lines [3]. Mobile clinics and telehealth have been used to increase healthcare access for isolated populations [4,5]; however, geographic limitations, like internet availability, are a barrier to mobile clinic implementation [6]. Satellite internet services are advertised to be capable of providing internet access to almost every corner of the globe, including internet speeds that allow uninterrupted services and smooth video consultations [7]. Most current information on the use of satellite internet in healthcare comes from news articles and industry reports, while the limited scholarly literature is outdated. For example, in the early 2000s, satellite internet supported a mobile wellness clinic in frontier and rural Montana [8] and, in 2010, satellite internet was utilized by a field hospital in earthquake disaster zones [9]. 

In the United States, health disparities are particularly acute among migrant and seasonal farmworkers, which are a predominantly rural, immigrant workforce critical to the nation’s agricultural economy [10]. This population faces unique health risks stemming from occupational exposures [11], language barriers [12,13], geographic isolation [14], immigration-related concerns, and limited access to healthcare due to lack of insurance [15]. The Farmworker Family Health Program (FWFHP) is a long-standing mobile clinic and collaboration that has been providing free healthcare to farmworkers who live and work in rural areas [16]. Delivering healthcare in these remote settings requires not only clinical expertise but also reliable, real-time data systems to document care and inform decision-making. While data collection is often viewed as secondary to direct patient care in resource-limited environments, the FWFHP has recognized that accurate data management is essential for improving health outcomes. As such, the program has prioritized strengthening its data infrastructure to better capture and respond to the unique health needs of this underserved workforce.

In 2019, the FWFHP staff implemented a relational database to collect electronic medical record (EMR) data for participants of the programs [17]. However, due to the internally perceived limitations of field-based data collection, the program was completing dual data entry, once on a paper form, and then finally into the REDCap database. The intervention, described here, was developed to overcome these data collection challenges. The aim of this methodological report is to describe the development, implementation, and evaluation of a real-time, live data-entry process using satellite internet in a rural mobile health clinic setting. Specifically, we seek to answer the following guiding question: Can satellite-enabled, real-time electronic medical record (EMR) data entry improve workflow efficiency and data accuracy in connectivity-challenged mobile clinic environments? We will also describe the equipment needed for direct data entry and provide an example of data tracking for the pharmacy services of the FWFHP mobile health clinic.

## 2. Materials and Methods (TIDieR Checklist 1-11)

The intervention will be described using the TIDieR checklist, a 12-item Template for Intervention Description and Replication [18]. TIDieR is an extension of the Consolidated Standards of Reporting Trials [19] and Standard Protocol Items statements [20] and was developed to improve the quality of reporting on interventions [21]. 

### 2.1. Why

Historically, data collection in rural public health endeavors has been completed using spreadsheets; the use of relational databases is a relatively new method for smaller public health projects [17]. For occupational health clinicians, this poses an interesting conundrum for ensuring data validity when working with rural workers in settings that may lack broader internet connectivity [22]. Real-time, live data entry is critical for syncing patient and medication data across multiple devices and maintaining the functionality of healthcare systems in remote or outdoor environments.

### 2.2. What

We planned and implemented real-time, live data entry using a satellite internet connection for a rural, mobile health clinic serving migrant farmworkers across southern Georgia, United States.

### 2.3. Materials

The technological infrastructure required for this project was a combination of both internet access technology and point-of-care data entry technology. The internet access technology consisted of Starlink© satellite internet [23]. Data-entry technology consisted of laptop computers, tablets, Bluetooth keyboards, a portable document scanner, barcode scanners, and an EMR developed prior to the intervention year [17]. A 240 Wh fully charged mobile battery was available to provide backup power supply for satellite internet and data-entry technology if needed. Additionally, mobile solar panels were used for on-site battery charging. A complete list of equipment, quantities, and locations within the mobile clinic can be found in Table 1.

### 2.4. Procedures

During the evenings of the 2-week-long program, medical clinics are set up on farms where the migrant workers both live and work and provide episodic medical care (e.g., acute care visits, physical therapy visits, and prescriptions provided by the pharmacy). From 2021 to 2023, all data were collected on paper and subsequently entered into the REDCap [24,25] as the EMR. A timeline of the data-entry procedures for the FWFHP can be found in Figure 1. This process was error-prone as there were often delays in data entry and a “mad dash” to ensure all data were entered before the conclusion of the 2 weeks, often on the last day of the program. To facilitate data collection and analysis, all instances of patient encounters within our database were systematically assigned a medical record number (MRN). Patients were assigned an MRN in the format of YYFXXX, with YY representing the current year, F representing a predetermined code for each farm (that matched the code label for the farm identification variable in REDCap), and XXX being a unique identifying number. This unique identifying number was assigned as farmworkers checked into the clinic and went in order from “001” to a possible “999”. To reduce transcription errors and improve data integrity, the REDCap system was programmed to prepopulate commonly used variables such as vital sign CPT codes and screening procedures with default values. All ICD-10 diagnosis codes were defaulted to “no” so that clinicians only needed to actively select “yes” for relevant diagnoses. Additionally, REDCap’s field validation features, including preset value ranges and required fields, were used to flag incomplete or implausible entries in real time, thereby supporting accurate and consistent data collection throughout the program.

#### 2.4.1. Data Security and Procedures

All patient data were collected and stored using REDCap, a secure and HIPAA-compliant data management platform designed for handling sensitive health information [24,25]. The REDCap system was hosted in a secure, restricted-access environment that met healthcare data protection standards. Access to the database was limited to authorized users through individual, password-protected accounts with role-based permissions. User authentication was managed through institutional login systems, and audit trails were maintained to document all data manipulation and export activities.

#### 2.4.2. Clinical Workflow

During mobile clinics for farmworkers, the first interaction with the patient occurs in triage, where upon consenting, vital signs and presenting problems are assessed and recorded by the nursing team. These data are used to triage patients as they move throughout the clinic to prioritize the care level with (1) a dental care provider, (2) a mental health provider, (3) a nurse practitioner, or (4) a physical therapist and so that patients are able to seek care at multiple stations as resources permit. Given that medications can be prescribed to a patient at multiple time points throughout the clinic, the final clinical station that patients pass through is the pharmacy. Farmworkers who have no presenting problem and only wish to have their vitals taken progress through the triage stations and then may leave the clinic. All stations are set up outdoors on the farm property, and the distance between each ranges from 10 feet to 200 feet. An overview of the clinical workflow of the FWFHP can be found in Figure 2. Data are input at each clinical station as workers make their way through the mobile clinic. Additionally, there is a need for data to be reviewed in “real time” as farmworkers make their way through the clinic, which was achieved through the implementation of this intervention.

### 2.5. Who Provided

The interprofessional program provides health screenings with student nurses, medical visits with student nurse practitioners, musculoskeletal evaluations with student physical therapists, oral health visits with student dental hygienists, mental health support with student social workers, and medication counseling and dispensing through a student-led mobile pharmacy. Allied health students participating in the program include pre-licensure nursing (i.e., students becoming registered nurses), post-licensure nursing (i.e., students becoming nurse practitioners), physical therapy, psychology, dental hygiene, and pharmacy; volunteers make up the Spanish–English interpreters and data management team. All clinicians and clinician trainees completed one-hour video-based REDCap training before working in the mobile clinics, followed by an additional one-hour in-person orientation focused on data-entry procedures and role expectations. During live data collection, on-site informatics support by the data management team was available and provided real-time troubleshooting and assistance to ensure smooth and accurate data entry.

### 2.6. How

The real-time data-entry Starlink© satellite internet made it possible to conduct real-time data entry, by having live Wi-Fi available on all iPads simultaneously. Starlink Roam is a satellite internet service offered by SpaceX [23], which uses low-Earth-orbit satellites to provide high-speed, low-latency internet access worldwide. The “roam” service specifically allows users to access the network while traveling, whether stationary or in motion.

### 2.7. Where

The FWFHP is based out of Colquitt County, GA, which is designated as rural by the Office of Management and Budgets [26]. This mobile clinic provides clinical care to approximately 600 migrant farmworkers and their families annually [16].

### 2.8. When and How Much

These intensive mobile clinics run for 2 weeks in June every year. This specific intervention with real-time data entry was first implemented in June 2023 and has been maintained since.

### 2.9. Tailoring

In the mobile pharmacy, medications were organized as “fast movers” or “non-fast movers”. “Fast movers” were identified based on historical dispensing of commonly used over-the-counter medications such as ibuprofen, acetaminophen, menthol-based muscle rub creams, multivitamins, and lubricant eye drops. “Fast movers” were prelabeled with generic directions for dispensing efficiency, and dispensing was logged using paper records due to limited QR scanners. After the conclusion of the program, the paper records for the “fast movers” were converted electronically into the REDCap system for accuracy. “Non-fast movers” required filling and dispensing by the mobile pharmacy. Filling included typing a prescription label in Spanish using label printers, dispensing the medication, and having a student pharmacist under the supervision of a licensed pharmacist double check for validity. Each “non-fast mover” prescription was scanned into REDCap using our drug- and dose-specific QR codes for the patient. Barcode scanners ensured a closed-loop system for medication administration by corresponding with the patient’s EMR [27]. 

The original FWFHP database was developed and first implemented in 2019 [17]. This process led to the identification of common diagnoses made by clinicians in the fields and increased capturing of procedure and evaluation and management (E&M) codes by placing them in the field-based EMR. REDCap allows for complete customization of data collection procedures, including variable coding (i.e., 0 = no and 1 = yes), variable naming, variable type (i.e., select-all-that-apply, single options, and free text), pre-selection of variable values, and branching logic to only allow for the selection of variables under specified criteria [24,25]. To facilitate a live-field-based data collection, we refined our original database from 2019 to prefill variable responses for values that were likely to not change. For example, Current Procedural Terminology codes for vital signs and health screening were defaulted as affirmative. International Classification of Diseases-10 codes were defaulted as negative, which meant clinicians needed only change the selection from “no” to “yes” for the diagnoses made. The database was also updated to reflect the NIH and Emory Nell Hodgson Woodruff School of Nursing’s common data elements [17,28], which had not been considered in the original design of the database.

### 2.10. Modifications

The intervention was further upgraded in 2024 by (1) adding 40 additional computer tablets so each clinician could connect to the internet and enter in data real time, and (2) adding a mesh Wi-Fi network to ensure adequate internet access across the mobile clinic.

### 2.11. How Well (Planned)

The fidelity of the intervention was assessed by data defects, defined as prescriptions that were not properly logged in REDCap, which led to missing prescriptions and uncertainty about whether a medication was dispensed to the patient. Data defects were calculated using the number of prescription discrepancies divided by the number of prescriptions entered into the data system, where discrepancies are variances in the number of prescriptions compared to what was recorded in the system. For example, if an individual had 4 total prescriptions listed in their chart, but only 2 were recorded or scanned into REDCap, this would be counted as 2 defects due to 2 missed/unknown prescriptions. REDCap’s built-in reporting tools were used to identify and quantify these discrepancies, allowing for efficient tracking of defect rates across the intervention period.

## 3. Results (TIDieR Checklist 12)

### How Well (Actual)

Out of the 319 prescriptions, only 6.9% (*n* = 22) were reported as data defects. Factors that may have contributed to data defects include user training, the data collection process, and the number of QR scanners. Pharmacy students rotated daily through numerous roles such as patient counseling, medication filling, label typing, or barcode scanning. Due to daily rotation, there was no consistency in roles; thus, there was the potential for a knowledge gap in appropriate barcode scanning. “Fast mover” data collection was handwritten and then manually converted into the REDCap system due to the high volume of “fast mover” dispensing. This two-step process may have resulted in human error due to difficulty reading handwriting or missing prescription data (omitted name, MRN, medication). Another factor that may have contributed to data defects was that the QR scanner was only used inside the mobile pharmacy to fill “non-fast mover” prescriptions.

## 4. Discussion

The rural setting of the FWFHP highlights the importance of reliable internet connectivity to support field-based healthcare tools. Wi-Fi extenders were instrumental in expanding network coverage to open areas, ensuring uninterrupted access to EMRs and other digital tools [29]. In the event of connectivity interruptions from storms, paper forms were used. Incorporating robust infrastructure, including backup power sources and offline-capable software, can further enhance resilience during adverse weather conditions [30]. While satellite internet users currently report inconsistent throughput and experience time delays, the technology is rapidly improving as more satellite internet providers enter the market [31]. Notably, SpaceX has announced plans to expand its Starlink network to a mega-constellation of 12,000–30,000 satellites aimed at delivering global satellite internet coverage [32]. As of December 2022, more than 3500 satellites had already been deployed, with over one million user terminals produced and service extended to at least 40 countries worldwide [33]. This rapid expansion of low-Earth-orbit satellite infrastructure holds promise for improving connectivity in even the most remote clinical environments.

The local success of this real-time data-entry intervention builds upon the foundational work of mobile health programs that have sought to address documentation challenges in rural and resource-limited settings. Traditionally, many mobile clinics have relied on paper-based data collection methods, with subsequent transcription into electronic systems post-visit. This asynchronous approach often introduces delays and increases the risk of transcription errors. For instance, Shahbodaghi et al. [34] highlighted the limitations of paper-based documentation, noting challenges in data accuracy and timeliness. They recommended standardizing the documentation process to reduce errors, which aligns with our intervention tailoring to include pre-determined variable coding, pre-selection of variable values, and application of branching logic to only allow for the selection of variables under specified criteria. Furthermore, seamless tracking of changes over time in the EMR is a capability particularly beneficial in coordinating care for migrant populations. In rural areas, adult farmworkers often access care individually based on their work schedules and geographic proximity to the clinic [12,13]. The inclusion of a farm identification variable and the current year in adult MRNs linked patients to specific locations. This allowed for streamlined care delivery for a geographically dispersed population when workers were referred to the brick-and-mortar migrant health clinic for further care. In addition to improving care coordination, this approach also establishes the foundation for longitudinal data collection and analysis, enabling the program to track health trends and outcomes over time across multiple locations and patient visits.

In the pharmacy component of the FWFHP, technology further enabled a mobile clinic team to provide streamlined and accurate care. We were able to link patients, prescriptions, and diagnosis codes, facilitating data analysis for quality improvement. This was previously not possible using Excel tracking. The new data management process improved the workflow and decreased the entry time into the REDCap system. The use of QR barcodes and scanner technology streamlined real-time data entry into REDCap, with a 6.9% defect rate. A recent systematic review and meta-analysis showed that across pharmacy settings, there is a 1.6% defect rate in prescription filling [35]. An important operational challenge observed during the implementation of real-time data entry was the variability introduced by daily staff and student rotations. Students and volunteers frequently moved between roles such as patient check-in, triage, medication filling, and barcode scanning, which meant that few individuals performed the same task consistently across multiple clinic sessions. While this rotation provided valuable educational experiences, it also introduced inconsistency in data-entry practices. Limited time for role-specific training and frequent role transitions tasks [36] likely contributed to the 6.9% defect rate observed in pharmacy data. Variability and lack of continuity are known risks in systems that rely on both electronic and paper-based documentation. Transitioning to a fully electronic system, supported by expanded access to barcode scanners, may help address these challenges by reducing reliance on handwritten records [27] and improving workflow standardization in rural settings. Future efforts should include structured, role-specific training [37] and daily refresher sessions [38] to promote data consistency between rotating staff and students and reduce workflow interruptions.

While the intervention demonstrated the feasibility of real-time data entry in connectivity-challenged rural settings, its long-term sustainability requires a careful consideration of ongoing costs, infrastructural maintenance, and operational scalability. The initial investment in satellite internet hardware, Wi-Fi mesh routers, tablets, barcode scanners, and backup power supplies represents a significant upfront cost for programs operating on limited budgets. The initial start-up budget for this project was USD 40,000, largely driven by the number of tablet computers needed. Ongoing subscription fees for the satellite internet service and the need to periodically replace or upgrade hardware further add to the financial burden. Additionally, the successful deployment of this model depends on the availability of trained personnel to manage both clinical care and data systems, as well as informatics support for troubleshooting in the field. Despite these challenges, the scalability of the intervention is promising. As satellite internet technology becomes more affordable and widely available, and as digital health tools continue to advance, this model could be adapted for use in other mobile health programs serving rural and underserved populations. Programs seeking to reduce costs may consider leveraging shared equipment across partner organizations, securing institutional or philanthropic funding for initial infrastructural investments, and negotiating group purchasing agreements or non-profit pricing for satellite internet subscriptions and hardware.

### Limitations

While this intervention demonstrated the feasibility of real-time data entry in a rural mobile health setting, several limitations should be noted. First, the data defect analysis was limited to pharmacy data and may not reflect potential errors in other clinical data fields. Second, staff and student rotations introduced variability in data-entry practices, which could have affected consistency. Additionally, this study was conducted in a single program over a limited two-week period, which may limit generalizability to other settings or larger-scale implementations. Future research should explore the impact of sustained use over multiple program cycles, evaluate the effectiveness of structured training protocols, and assess cost-effectiveness compared to other data collection models. Expanding this model to other mobile clinics and documenting implementation outcomes across diverse geographic regions will be important next steps to validate the scalability and long-term sustainability.

## 5. Conclusions

The implementation of real-time, live data entry using satellite internet and integrated digital tools represents a scalable and effective model for enhancing healthcare delivery in connectivity-challenged rural settings. By leveraging a customized REDCap database, barcode scanners, Wi-Fi extenders, and backup power sources, the initiative successfully addressed the limitations of traditional paper-based documentation, reducing errors and streamlining clinical workflows. For mobile clinics operating in rural areas with inconsistent or no internet access, this approach offers a transformative solution. By reducing the reliance on paper records and improving the accuracy of patient data collection, mobile teams can ensure a continuity of care, particularly for migrant populations who require seamless health record tracking across multiple visits and locations. Expanding this model to other mobile clinics could significantly enhance access to care in rural and underserved communities. As satellite internet technology advances and digital tools become more widely available, mobile clinics can achieve greater efficiency, improve patient outcomes, and strengthen healthcare equity by ensuring that even the most geographically isolated populations receive high-quality, data-driven care.

## Figures and Tables

**Figure 1 ijerph-22-00842-f001:**
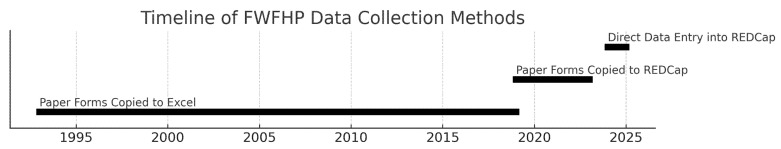
Timeline of FWFHP data collection methods.

**Figure 2 ijerph-22-00842-f002:**
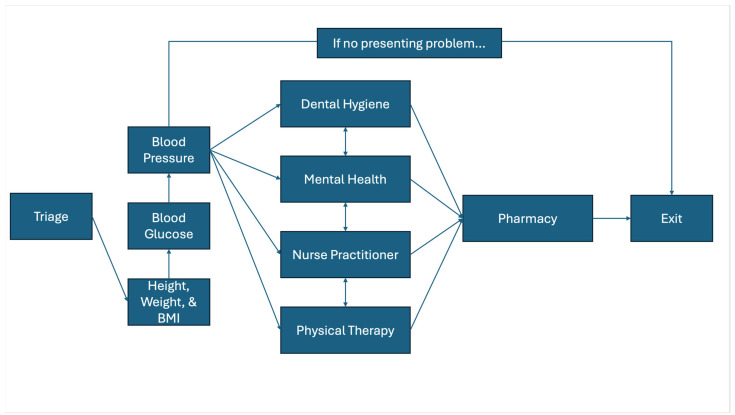
Clinical workflow of the FWFHP.

**Table 1 ijerph-22-00842-t001:** List of technology infrastructure required, quantity, and approximate location to facilitate real-time, field-based data entry.

Internet Access Technology		
**Equipment**	**Quantity**	**Approximate Location**
Satellite	1	Unobstructed view of sky
Mobile Base	1	Unobstructed view of sky
Internet Modem	1	Unobstructed view of sky
Starlink© Cable (50 ft)	1	Connects modem and satellite
AC Power Cable (6 ft)	1	AC power to Starlink
Extension Cords (50 ft)	2	AC power to mesh routers
Wi-Fi Mesh Network Routers ^#^	4	1 at triage, nurse practitioner treatment area, physical therapy treatment area, and pharmacy
Battery Packs	5	1 powering internet modem and 1 powering each of the Wi-Fi mesh network routers. In areas without electricity
**Point-of-Care Data-Entry Technology**		
**Equipment**	**Quantity**	**Approximate Location**
Laptop Computers	4	2 pharmacy, 2 data managers: mobile
Tablet Computers with Electronic Pencil *	4	Check-in
	20	Triage, screening stations
	2	Check-out
Tablet Computers with Bluetooth Keyboards *	15	Nurse practitioner treatment area
	10	Physical therapy treatment area
	2	Mental health treatment area
	5	Clinic preceptors
Portable Document Scanner	1	Pharmacy
Barcode Scanner	2	Pharmacy

* These numbers were increased in 2024 through the addition of 40 extra tablets that could be used for this intervention. ^#^ Wi-Fi mesh routers were added when the intervention was modified in 2024.

## Data Availability

Data are available upon reasonable request from the corresponding author.

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
