# Peer review of "Improving Rural Healthcare in Mobile Clinics: Real-Time, Live Data Entry into the Electronic Medical Record Using a Satellite Internet Connection"

_ijerph, 2025, doi:10.3390/ijerph22060842_

Round 1

Reviewer 1 Report

Comments and Suggestions for Authors

The topic proposed by the researchers is important and current. However, a serious problem is the extremely laconic introduction. The article lacks a foundation in existing scientific literature. The poor bibliography suggests that the authors did not conduct appropriate literature studies. Even taking into account the methodological nature of the article, it is necessary to show the researchers' previous approaches in similar works. It is also possible to show some elements of bibliometric analysis.

I appreciate the presentation of materials and methods. This is definitely the strongest part of this work, which is in line with the assumption. Despite the nature of the article, the part concerning the results is insufficient and, even for this type of work, definitely too poor.

To sum up, a methodological article must have a strong foundation in previous research and also result from the achievements of other researchers. Showing different approaches to methodology by other authors is necessary to be able to consider the presented methodology as valuable.

Author Response

Please see attachment for all reviewers. 

Reviewer 2 Report

Comments and Suggestions for Authors

The study likely addresses a significant problem in its domain, contributing to existing knowledge. But the methodology has to be well-structured and justified. It is better to define the method and experimental results. 

Author Response

Please, see attachment for all reviewers. 

Reviewer 3 Report

Comments and Suggestions for Authors

The manuscript presents a valuable, innovative application of satellite internet and digital tools to improve healthcare delivery in rural mobile clinics. The use of the TIDieR checklist to structure the methodology is commendable and improves replicability. The authors clearly describe the intervention, setting, and outcomes, which include a low rate of data defects and enhanced clinical workflow.

That said, a few areas could be improved:

1. Introduction: Clarify the specific research aim or guiding question earlier in the abstract and introduction to frame the paper more clearly as a methodology report with evaluative elements.

2. Sustainability: Consider briefly discussing the long-term sustainability of the intervention, including equipment costs, ongoing internet subscription, and training needs.

3. Data Quality Discussion: While a 6.9% defect rate is presented, adding comparison with standard benchmarks (e.g., pharmacy data entry error rates) would help contextualize this performance.

4. Training and Human Factors: Expand the discussion of staff/student training, particularly addressing rotation and its potential impact on data consistency.

5. Visual Aid: A diagram or flowchart summarizing the mobile clinic workflow and data flow could greatly enhance clarity for readers unfamiliar with the program.

Overall, this is a strong and well-documented piece of work. With minor improvements in presentation and contextualization, it will be an important contribution to public health technology literature.

Author Response

(The authors gave the same response as above.)

Reviewer 4 Report

Comments and Suggestions for Authors

Dear Authors,

thanks a lot for the possibility to read your manuscript.

In the following, I will try to provide some suggestions to improve the work.

Good luck for the final publication!

Introduction

  • The Introduction is very simple and short. I suggest to better define the rationale of the research as well as many limitations or gaps currently existing in the literature to highlight the necessity and the value of the work.
  • Although the Introduction mentions the geographical difficulties and inequalities in access to health care, it could benefit from further discussion of the theoretical background to the use of mobile and satellite technologies in rural health care. For example, a brief discussion of previous applications or similar research could better contextualise the innovative nature of the intervention described. As well, some information regarding previous research conducted with similar objectives should be added to enhance the positioning of the study in comparison with the extant literature or other practical interventions.
  • At the end of the Introduction, a clear definition of the researchh question(s).

Materials and Methods

This section is very clear. However, I suggest:

  • to better describe how the methodological approach chosen is useful to address the defined research questions or objectives,
  • to better clarify how the results were evaluated would be helpful. For example, how were the data collected through the REDCap system analysed? How were errors in the data monitored?
  • to detail the data collection and the data management process, considering how sensitive data is handled as well as data security and privacy guaranteed, especially considering the importance and relevance of these topics within the healthcare context.

Results

The Results’ section could benefit from a visual representation of the data (graphs or tables) to facilitate understanding and comparison.

Discussion and conclusion

These two sections should benefit from the definition of a more critical analysis comparing the proposed intervention with other research or practical previous activities, as well as some reflections on the limitations of the intervention. In addition, the discussion could deepen the long-term sustainability of the intervention. Has the cost of the technological infrastructure (satellite, Wi-Fi extenders, mobile devices) been considered? What are the prospects for the expansion of this model in other areas or for its maintenance?

Moreover, these two sections might be enhanced adding the contributions of the study, both from a theoretical and a practical perspective as well as detailing some directions for future research. For example, exploring further improvements in technologies to reduce the defect rate or testing the system in other locations or with other vulnerable populations.

Author Response

Please, see attachment to all reviewers. 

Reviewer 5 Report

Comments and Suggestions for Authors

well designed well described informative communication 

Author Response

(The authors gave the same response as above.)

Round 2

Reviewer 1 Report

Comments and Suggestions for Authors

I appreciate the changes introduced by the authors, especially the significant expansion of the sources used in the work, the development of the discussion and the description of identified research limitations, which is evidence of scientific maturity.

Reviewer 2 Report

Comments and Suggestions for Authors

Now it's more clear, so it may be accepted in its present form.